# Recurrent SARS-CoV-2 infections and their potential risk to public health – a systematic review

Seth Kofi Abrokwa[1], Sophie Alice Müller[2], Alba Méndez-Brito[1], Johanna Hanefeld[2], Charbel El Bcheraoui[1]*

1 Evidence-based Public Health, Centre for International Health Protection, Robert Koch Institute, Berlin, Germany, 2 Centre for International Health Protection, Robert Koch Institute, Berlin, Germany

☯ These authors contributed equally to this work.
* El-BcheraouiC@rki.de

**Data Availability Statement:** All relevant data are within the paper and its Supporting Information files.

## Abstract

### Objective

To inform quarantine and contact-tracing policies concerning re-positive cases—cases testing positive among those recovered.

### Materials and methods

We systematically reviewed and appraised relevant literature from PubMed and Embase for the extent of re-positive cases and their epidemiological characteristics.

### Results

In 90 case reports/series, a total of 276 re-positive cases were found. Among confirmed reinfections, 50% occurred within 90 days from recovery. Four reports related onward transmission. In thirty-five observational studies, rate of re-positives ranged from zero to 50% with no onward transmissions reported. In eight reviews, pooled recurrence rate ranged from 12% to 17.7%. Probability of re-positive increased with several factors.

### Conclusion

Recurrence of a positive SARS-CoV-2 test is commonly reported within the first weeks following recovery from a first infection.

## Introduction

After 14 months in the COVID-19 pandemic, health systems worldwide have still not achieved control of the severe acute respiratory syndrome coronavirus-2 (SARS-CoV-2). SARS-CoV-2 is highly transmissible with a potential secondary attack rate of more than 17% [1]. Further, this rate of transmission has been reported to be even higher in circulating variants of concern

**Funding:** The authors received no specific funding for this work.

**Competing interests:** The authors have declared that no competing interests exist.

such as the B.1.1.7 than in pre-existing variants [2]. Since its emergence in December 2019, the SARS-CoV-2 virus has infected more than 120 million people and led to at least 2.6 million deaths globally [3]. In addition to the high disease burden, the virus has brought an unprecedented downpour of social and economic setbacks, the course of which cannot start to be reversed until herd immunity, natural or artificial, is achieved. While six vaccines are already licensed, we are still far from herd immunity, given that vaccines need to be produced at scale, priced affordably, and allocated globally to be widely deployed [4,5]. Additionally, in consideration of emerging variants and reports of recurrent SARS-CoV-2 infections, the global battle against the virus is far from being over.

Immunological evidence suggests that immune response to post-natural SARS-CoV-2 infection is transient with reported rapid depletion of antibodies in the first four months followed by gradual waning within a year [6–9]. Recent studies have detected viral nucleic acid in previously recovered patients [10] and the first symptomatic reinfection with different viral strains has been confirmed in June 2020 in North America [11]. In the near future, this risk for reinfection is expected to increase due to lack of protective immunity and circulation of new variants. Whereas some studies did not find replication-competent virus in re-positives—a positive test following recovery from a first SARS-COV-2 infection—[12,13], a study in Spain reported onwards transmission [14]. This potential for onwards transmission is of utmost importance as it increases the diseases burden of SARS-CoV-2 and its associated complications. This risk is even higher as previously infected people tend to adhere less to mitigation measures, such as social distancing and public health policies [15]. Adding to the danger of behavioral factors, currently there are no harmonized protocols for contact tracing or isolation for re-positives globally.

Public health policies concerning re-infections vary globally as evidence on the extent and the potential of the consequences is lacking. The European Center for Disease Control (ECDC) recently reported that reinfection with SARS-CoV-2 remains a rare event [16]. They further stated that there is the risk for some re-infected persons to transmit SARS-CoV-2 infection to susceptible contacts as previous infections do not produce sterilizing immunity in all individuals. However, there is insufficient evidence to determine the effect of previous infection on the risk of onward transmission and its impact on public health safety. This is further complicated by the limited evidence on how newly circulating variants of concern affect the probability of reinfection and their role in onward transmission. Currently, COVID-19 mitigation policies in the United States do not recommend retesting or quarantine if a previously recovered patient is exposed to SARS-CoV-2 within 90 days after the date of symptom onset from the initial SARS-CoV-2 infection [17]. In the EU and Germany, public health authorities only impose quaratine in the first 90 days if the person works or lives with a risk group [18] and/or a variant of concern is suspected [19]. If the second exposure occurs more than three months after the first infection, the previously recovered person is considered at the same risk as any other contact person without previous infection [18]. As more recurrent SARS-CoV-2 infections are being reported, whether such episodes are actual reinfections or not, constant evidence on the cause of re-positives and onwards spread is crucial to inform public health policies. We reviewed and synthesized the evidence around the extent and characteristics of reinfections and re-positive rates of SARS-CoV-2, to inform related quarantine and contact-tracing policies.

## Materials and methods

We conducted a review of scientific literature following the Preferred Reporting Items for Systematic Review and Meta-analysis (PRISMA) [20]. We performed the literature search using

the electronic databases PubMed, Embase and preprint servers (ArRvix, BioRvix, ChemRvix, MedRvix, Preprints.org, ResearchSquare and SSRN). We used novel coronavirus search terms developed by the Robert Koch Institute library, and terms for reinfection, re-positive, reactivation, relapse, recurrence, and secondary infection. The S1 Table provides the detailed search strategy. The search was restricted to SARS-CoV-2 infection in humans, to publication date since 2020 and to English language. The study was exempt from institutional review board approval because no primary data was included. The review protocol has not been previously registered or published.

We organized the search results and removed duplicates using Endnote X7 (Clarivate Analytics) [21]. Title and abstract screening of publications were conducted using Rayyan QCRI web application for systematic reviews [22]. Case reports, case series, observational studies and reviews were included. Publications reporting on cases of one-episode of SARS-CoV-2 infection, manuscripts without primary data (letter to the editor, conference abstract, commentaries), model stimulation studies, laboratory studies, and animal models were excluded. One reviewer (CEB) developed the data extraction forms and two reviewers (SKA and SM) extracted data of the included publications. To minimize potential errors in the content, each reviewer examined a random 20% of data extracted by the other.

The following data were extracted for all included publications: first author, year, country, age study population, number of cases, testing methods, symptomatology, duration between infections and onward transmission of infection. Additional data extracted for the specific study designs included: 1) case reports: comorbidities, reinfection confirmation, infection differentiation method, relaxation of protective behavior, symptomatology of onward infection, 2) observational studies: publication status, country and setting, serology test, evidence for reinfection, total sample size, risk of reinfection, incidence rate, whether the re-positive was identified due to symptoms or not and 3) reviews: publication status, types of studies reviewed, reinfection confirmation, reported association with demographics, necessity of intensive care treatment, and comorbidities.

Three independent reviewers (SKA, SM, AMB) evaluated the quality and risk of bias of included publications. We assessed the publications using Joanna Briggs Institute (JBI) critical appraisal tool for case series and evidence review, and adapted the JBI critical appraisal tool for case reports [23]. For observational cross-sectional and cohort studies, we adapted the National Heart, Lung and Blood institutes protocol [24]. Data from included publications were analyzed descriptively focusing on re-positive rates, epidemiological characteristics of recurrent SARS-CoV-2 infection and onward transmission. The relevant extracted data were organized and presented in tables.

## Results

The primary database search on February 2, 2021 yielded 2,736 publications. After removing duplicates, and performing title and abstract screening, we retrieved 199 publications for full text screening (Fig 1).

After applying the eligibility criteria in the full text assessment, 133 publications were included. The included publications comprised 75 case reports, 15 case series, 35 observational studies and 8 reviews. This includes two additional manuscripts published after the initial database research but deemed essential given the relevance of their content [25,26].

In case reports and case series, a total of 276 cases between the age of three [27] and 93 [28] years were recorded as re-positive by PCR. About 101 cases had one or more comorbidities. The characteristics of cases are displayed in the S2 Table. The duration between the two infection episodes varied between one [29–33] and 32 weeks [34]. Applying the Robert Koch

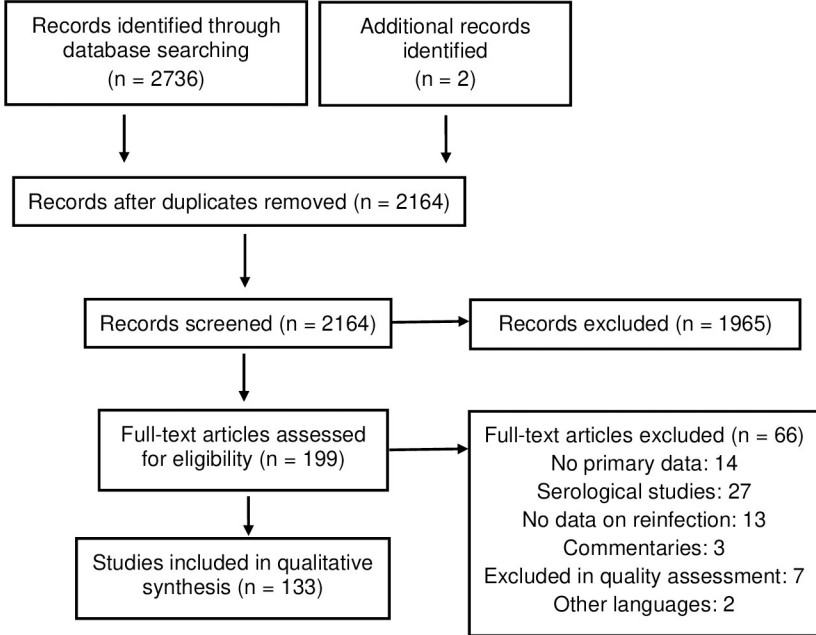

**Fig 1. Selection of studies.**

Institute's (RKI) definition for probability of reinfection [35], 217 non-previously confirmed reinfection cases were classified as possible reinfection and eight case as probable reinfection [27,36–41] as shown in S2 Table. Thirty-eight cases were not classified as either possible, probable or confirmed reinfections as not all case definition elements were reported as required by RKI criteria. Twelve cases were confirmed to be reinfections through whole genome sequencing of viral material in both episodes [11,42–50]. Fifty percent of all confirmed reinfections were reported to have occurred within 90 days after the initial disease. Phylogenetic analysis in one re-positive case sample, identified new viral strain which was absent in the location of exposure during patient's first episode [14]. As this finding did not meet the RKI criteria, this re-positivity could not be classified as confirmed reinfection. Clinical characteristics of confirmed reinfections and the re-positive case with new viral strain are detailed in Table 1. Seven studies, including one study of confirmed reinfection, reported follow-up testing of contacts of re-positives [14,39,47,51–54]. Four of these studies identified positive contacts. The positive contacts from three studies included family members in two studies [14,55], and one treating physician in another study [51]. In the fourth study, viral genomic materials were identified to be identical in a re-infected health care worker (HCW) and three patients. The clustered nature of the transmission suggested a possible index case, however as symptoms of COVID-19 infection was first observed in a patient who received no care from re-infected HCW, it was unclear whether the re-infected HCW was the index patient [47]. The quality assessment of case reports and case series were rated on a scale of zero to nine and ten respectively as shown S3 and S4 Tables. Case reports with evidence of confirmed reinfections were among the highly rated publications. The top-rated case series reported re-positivity occurring between three to six weeks after initial disease [56,57]. One of the top-rated case series reported high rate of re-positivity to be associated with younger age, low body mass index and moderate disease severity [57].

The 35 observational studies, including four preprints [61–64], were predominantly conducted in China and focused on healthcare settings as shown in Table 2. A total of 1,100 re-

**Table 1. Characteristics of confirmed SARS-COV-2 reinfection cases.**

| Author | Country | Number of cases | Age in years | Co-morbidities | Symptoms at 1st episode | Time (weeks) between 1st and 2nd episode | Symptoms at 2nd episode | Onward transmission from 2nd episode |
|---|---|---|---|---|---|---|---|---|
| Tillett RL et al. [11] | USA | 1 | 25 | No | Sore throat, cough, headache, nausea, diarrhoea | 6 | Fever, headache, dizziness, cough, nausea, diarrhoea, shortness of breath | NR |
| Mulder M et al. [58] | Netherlands | 1 | 89 | Yes | Fever, severe cough | 8 | Fever, cough, dyspnoea, tachypnoea | NR |
| Prado-Vivar B et al. [59] | Ecuador | 1 | 46 | No | Headache, drowsiness | 9 | Odynophagia, nasal congestion, fever, back pain, productive cough, dyspnoea | NR |
| To KK et al. [49] | Hong Kong | 1 | 33 | No | Productive cough, sore throat, fever, headache | 10 | Asymptomatic | NR |
| Van Elslande J et al. [48] | Belgium | 1 | 51 | Yes | Headache, fever, myalgia, cough, chest pain, dyspnoea, anosmia, change of taste | 12 | Headache, cough, fatigue | NR |
| Colson P et al. [50] | France | 1 | 70 | Yes | Fever, cough | 15 | Asymptomatic | NR |
| Goldman JD et al. [42] | USA | 1 | 60–69 | Yes | Fever, chills, productive cough, dyspnoea, chest pain | 20 | Dyspnoea, dry cough, weakness | NR |
| Harrington D et al. [34] | UK | 1 | 78 | Yes | Fever | 32 | Shortness of breath | NR |
| Lee JS et al. [41] | South Korea | 1 | 21 | No | Sore throat, cough | 3 | Sore throat, cough | NR |
| Gupta V et al. [43] | India | 2 | 25–28 | No | Asymptomatic | 9–14 | Asymptomatic | NR |
| Selhorst P et al. [60] | Belgium | 1 | 39 | NR | Cough, dyspnoea, headache, fever, general malaise | 26.5 | Mild symptoms | Unclear |
| Pérez-Lago L et al* [14] | Spain | 1 | 53 | Yes | Dyspnoea, fever, cough | 20 | Respiratory failure | Yes |

* Phylogenetically confirmed but does not meet RKI confirmed reinfection criteria,

NR not reported.

positives were reported out of 180,185 previously recovered patients, whereby one study reported 44 re-positives of an unknown total assessed [65]. Reported re-positive rates ranged from zero to 50% [66,67] in patients aged between two months [68] and 95 years [69]. At least 40% of re-positives were found to be symptomatic at the second episode (451/1,046). The duration between discharge/negative test/completion of therapy and re-positivity varied from less than one [12] to 33 weeks [62]. Highest re-positive rates of more than 20% were reported to occur in a follow- up period between one to seven weeks [64,66,70,71], while low re-positive rates occurred in a follow-up period of more than nine weeks [25,62,63,67]. Only two studies performed genome sequencing from naso-/oropharyngeal samples, but full-length viral genomes could not be obtained [12,72]. Eight studies reported testing or follow-up of contacts, but no onward transmission was identified [70,72–78]. The quality of observational studies was assessed on a scale of zero to 14. Included publications were rated between 3 [79] and 11 [65,78,80], as shown in S5 Table. Two of the three top rated publications reported re-positive rates of 6.25% within five weeks [80] and 19.81% within three to five weeks [78] after initial infection. The third top-rated study reported 44 re-positive cases two weeks post-discharge [65]. The two studies, that included healthcare workers were scored 10/14 and had one of the lowest re-positive rates of 0% and 0.32% [62,67].

**Table 2. Characteristics of confirmed SARS-COV-2 reinfection cases as reported in observational studies analysed.**

| Author | Preprint | Country and Setting | Total | Reinfection cases | Age in years | Re-positive rate (%) | Time in weeks between 1st and 2nd episode | Symptomatic at reinfection (%) | Onward transmission from 2nd episode |
|---|---|---|---|---|---|---|---|---|---|
| Hanrath AT et al. [67] | No | UK, healthcare workers | 1038 | 0 | NR | 0.00% | 24 | NA | NR |
| Abu-Raddad LJA et al. [63] | Yes | Qatar, surveillance | 133266 | 54 | median 33 range (16–57) | 0.04% | median 9.3 (6.4–18.4) | 42.6% | NR |
| Pilz et al. [25] | No | Austria, surveillance | 14840 | 40 | median 39.8 range (26–55) | 0.27% | 30±4 | 7.5% hospitalisation | NR |
| Lumley SF et al. [62] | Yes | UK, healthcare workers | 1246 | 4 | NR | 0.32% | 22.8–33 | 25.0% | NR |
| Patwardhan A [68] | No | USA, children | 989 | 4 | median 3.55 range (0.2–13) | 0.40% | 1–3 after last negative | 50.0% | NR |
| Hansen CH et al. [26] | No | Denmark, surveillance | 11068 | 72 | NR | 0.65% | >12 | NR | NR |
| Luo S et al. [81] | No | China, hospital patients | 1673 | 13 | NR | 0.78% | NR | 100% | NR |
| Pan L et al. [73] | No | China, hospital patients | 1350 | 14 | 44.4 ± 15 | 1.04% | 1.7 ± 0.7 after discharge | 7.1% | not found |
| Kang YJ et al. [82] | No | South Korea | 7829 | 163 | in groups | 2.08% | >1–5 after discharge | 43.9% mild | NR |
| Du HW et al. [74] | No | China, hospital patients | 126 | 3 | NR | 2.38% | 1–3 after treatment | >66.7% | not found |
| Ali AM et al. [61] | Yes | Iraq, hospital patients | 829 | 26 | range 10–60 | 3.14% | 3.7–19.7 after recovery | 96.2% | NR |
| Wang X et al. [83] | No | China, hospital patients | 193 | 12 | 55.5 ±13.7 | 6.22% | 4–8 | 83.3% | NR |
| Zhou J et al. [80] | No | China, hospital patients | 368 | 23 | 51±16 | 6.25% | 5 | 82.1% | NR |
| Qiao XM et al. [75] | No | China, hospital patients | 15 | 1 | NR | 6.67% | 2 | 100% | not found |
| Hu J et al. [84] | No | China, hospital patients | 117 | 8 | 46.25 ± 17.7 | 6.84% | median 1.8 (1.7–2.3) after discharge | 100% | NR |
| Tao W et al. [85] | No | China, hospital patients | 173 | 12 | NR | 6.94% | 5 | 0% | NR |
| Liu T et al. [86] | No | China, hospital patients | 150 | 11 | median 49 range (37–62) | 7.33% | 2 | NR | NR |
| Chen J et al. [87] | No | China, hospital patients | 1087 | 81 | median 62 range (16–90) | 7.45% | median 1.3 (0.4–2.6) | 84%mild; 16% moderate/severe | NR |
| Zheng J et al. [88] | No | China, hospital patients | 285 | 27 | median 44 range (32–62) | 9.47% | 1 (0.7–1.1) | 37.1% | NR |
| Bongiovanni M et al. [69] | No | Italy, hospital patients | 1146 | 125 | mean 65.7 95% CI (26–95) | 10.91% | 2.9 (95% CI 0.4–6.1) after discharge | 23.2% | NR |
| Zhang K et al. [89] | No | China, hospital patients | 220 | 27 | 22–78 | 12.27% | 1–8 | 22% mild; 77% moderate | NR |
| Tian M et al. [90] | No | China, hospital patients | 147 | 20 | mean 37.2 range (4–80) | 13.61% | 1–6.7 after discharge | 0% | NR |
| Lu J et al. [12] | No | China, hospital patients | 619 | 87 | median 28 range (0.3–69) | 14.05% | 1 (0.3–2.7) after discharge | 0% | NR |
| An J et al. [76] | No | China, hospital patients | 262 | 38 | 94.7% < 60y | 14.50% | 2 | 29% mild; 71% moderate | not found |
| Yuan J et al. [91] | No | China, hospital patients | 172 | 25 | median 28 range 16.3–42 | 14.53% | 1 week ± 0.55 after last negative | 32% mild | NR |
| Wu J et al. [92] | No | China, hospital patients | 60 | 10 | NR | 16.67% | 1–8 | 20% mild | NR |

*(Continued)*

**Table 2.** (Continued)

| Author | Preprint | Country and Setting | Total | Reinfection cases | Age in years | Re-positive rate (%) | Time in weeks between 1st and 2nd episode | Symptomatic at reinfection (%) | Onward transmission from 2nd episode |
|---|---|---|---|---|---|---|---|---|---|
| Yang C et al. [72] | No | China, hospital patients | 479 | 93 | median 34 95% CI (29–38) | 19.42% | 2–6 | 28% mild | not found |
| Abdullah MS et al. [77] | No | China, hospital patients | 138 | 27 | 41.3 ± 17.0 | 19.57% | 1.5 after discharge | 22% mild | not found |
| Wong J et al. [78] | No | Brunei, hospital patients | 106 | 21 | median 47 | 19.81% | 3–5 | 5% | not found |
| Xiao AT [64] | Yes | China, hospital patients | 70 | 15 | median 64 range (51–73) | 21.43% | 3–7 | NR | NR |
| Li Y et al. [71] | No | China, hospital patients | 13 | 4 | median 37 range (1–73) | 30.77% | 0.7–2 after discharge | 100% | NR |
| Zhang X et al. [93] | No | China, hospital patients | 59 | 19 | NR | 32.20% | NR | 0% | NR |
| Peng D et al. [70] | No | China, hospital patients | 38 | 14 | 7.2±4.8 | 36.84% | 1–5 | 71.4% | not found |
| Zhao W et al. [66] | No | China, hospital patients | 14 | 7 | median 5.7 range (3–7) | 50.00% | median 2 (1–2.4) after discharge | 14.3% | NR |
| Chen LZ et al. [65] | No | China, hospital patients | NA | 44 | 49.68 ± 16.80 | NA | 2 after discharge | >63.6% | NR |

NR not reported, CI confidence interval.

Table 3 details the eight selected literature reviews including five meta-analyses. The quality and risk of bias assessment of the reviews ranked between four and 11 on a scale of 11 as shown in S6 Table. The largest review included 85 publications and primary data on nine patients [94]. In this review, a total of 1,350 re-positive cases were identified, and a mean duration of re-positivity of 34.5 days after initial infection was observed in 123 cases, that were confirmed recovered after two negative PCR tests. Two high-rated reviews reported pooled recurrence rate of SARS-CoV-2 to be 14.6% (95%CI: 11.1–18.1%) [95] and 17.7% (95% CI: 12.4%-25.2%) [96]. Regarding duration between initial and recurrent episode, the top-rated study reported pooled estimate of the interval of disease onset to recurrence to be 35.4 days (95% CI 32.65–38.24 days) and the pooled estimate from last negative test to recurrence of infection to be 9.8 days (95% CI 7.31–12.22 days) [95]. Additionally, the time from discharge to recurrence of SARS-CoV-2 was reported by two other high-rated reviews to be 13.4 days (12.1–14.7) [96], and between two and 22 days. No cause of re-positivity was identified in the reviews, although the probability of recurrent SARS-CoV-2 infection increased with prolonged initial illness, moderate disease severity, decreased leucocytes, low platelets and low CD4 count [96]. The association of recurrent SARS-CoV-2 infection with age was controversial. Both, young and old age were identified as risk factors for recurrent SARS-CoV-2 infection [95,96]. None of the reviews reported on onwards transmission. Within the reviewed studies, in a report from Korea CDC where 790 contacts of 285 re-positive cases were monitored, no case was identified as newly infected from contact with re-positive cases during the re-positive period [97].

## Discussion

To the best of our knowledge, this is the first literature review rating evidence on more than 1349 recurrent SARS-CoV-2 infections worldwide. In this review, we found that recurrence of a positive SARS-CoV-2 test among previously recovered patients is common. Some re-positives follow exposure and/or present severe illness including death. The incidence of re-

**Table 3. Characteristics of confirmed SARS-COV-2 reinfection cases as reported in reviews analysed.**

| Author | Preprint | Studies reviewed | Total studies | New positive respiratory samples after recovery | Duration from symptoms to $1^{st}$ +, $1^{st}$ -, $2^{nd}$ +, 2nd– (days) | % cases with ≥1 comorbidity | Associated demographics | Cases requiring admission to ICU | Confirmed reinfection | Onward transmission from $2^{nd}$ episode |
|---|---|---|---|---|---|---|---|---|---|---|
| Arafkas M et al. [98] | No | Literature review + meta-analysis | 7 (3 case reports, 1 case series, 2 clinical studies, 1 in-vivo study) | 15 + 9 persistent positive | NR | NR | None | 6 deaths | No | NR |
| Azam M et al. [95] | No | Systematic review + meta-analysis | 14 (8 cohort, 6 cross-sectional) | 14.6% (95% CI 11.1–18.1%) | 35.4 (95% CI 32.65–38.24) 9.8 (95% CI 7.31–12.22) | NR | Risk factor: young age, long initial illness; protective factor: diabetes, severe disease, low lymphocyte | NR | No | NR |
| Dao TL et al. [97] | No | Narrative review | 62 | Few hundred | NR | 1 study: 64% cases with comorbidities | None | 3 | 1 study | Not found |
| Elsayed SM et al. [99] | No | Systematic review | 11 case reports | 11 | 1st negative to 2nd positive: 2–22 | NR | NR | NR | NR | NR |
| Gidari A et al. [94] | No | Systematic review + primary data | 85 (32 case reports, 50 case series, 5 reviews) | 1341 + 9 primary data | 6.2(4.7), 19.1 (10.2), 34.5 (18.7), 41.2 (21.5) | 34.5% | None | 2 ICU cases | No | NR |
| Hoang T et al. [100] | Yes | Literature review + meta-analysis | 37 (14 case reports, 5 case series, 18 observational studies) | 16% (95%CI 12–20) | Disease onset to admission 17.3, admission to discharge 16.7, discharge to re-positive 10.5 | 43% (95% CI 31–55) | NR | NR | No | NR |
| Mattiuzzi C et al. [101] | No | Literature review + meta-analysis | 17 clinical studies | 12% (95%CI 12–13) | Discharge to re-positive 1–60 | NR | NR | NR | No | NR |
| Yao MQ et al. [96] | No | Systematic review + meta-analysis | 10 | 17.7% (95% CI: 12.4%-25.2%) | Discharge to re-positive 13.38 (95% CI: 12.08–14.69) | NR | Age, moderate severity, bilateral pulmonary infiltration, low leucocytes/ platelets/CD4 | NR | No | NR |

NR not reported, CI confidence interval, ICU intensive care unit.

positivity varied with duration after initial disease, high rates of re-positivity with pooled incidence of 12.0% to 17.7% was observed within the first 90 days after initial infection as compared to rates (less than 1%) after 90 days or more. High re-positive rates within 3 months after initial infection raise questions on the cause of re-positivity, such as potential of prolonged viral shedding, testing errors or actual re-infections. Certainly, the high rates of re-positivity are of concern for current COVID-19 public health policies.

Current international policies base their recommendations for contract tracing and travel restrictions on a duration of 90 days after initial infection [17,18]. The US Centers for Disease

Prevention and Control (CDC) does not impose quarantine measures within 90 days of re-exposure. The CDC regards re-positivity within 90 days of re-exposure more likely as persistent shedding of viral RNA than reinfection and states that the risk of potential SARS-CoV-2 transmission are likely outweighed by the personal and societal benefits of avoiding unnecessary quarantine [17]. The ECDC also identified in their assessment that ongoing vaccine trials have been focused mainly on their efficacy and effectiveness in reducing disease outcome such as severity of disease or induced mortality and not on their ability to reduce the risk of SARS-CoV-2 transmission from infected vaccinated individuals to susceptible contacts [16]. The ECDC therefore underlines the need for follow-up studies to better assess the potency and duration of protection from reinfection and their effect against further transmission of contacts [16].

Whole genome sequencing is key to identifying the causes for re-positivity. However, confirming re-infections through genome sequencing is rarely performed given the difficulty in ascertaining the first infection in the absence of stored genetic material and given the large number of infected people worldwide [3]. Only 15 out of 124 included publications with primary data reported on whole genome sequencing.

In the absence of a validated re-infection definition, often a clinical perspective is applied. By applying the clinical definition of RKI, the majority of re-positives in our review were classified as possible reinfections and only 12 were confirmed reinfections as stated in the original studies. Notwithstanding the limited number of confirmed reinfections, we showed in the present review that 50% of genetically confirmed cases of re-infection were observed within 90 days after initial infection. This finding questions the current recommendation on contract tracing and travel restrictions. In view of our results, application of the current regulations could lead to an underestimation of re-infections and their potential threat to public health measures. There is therefore the need to continuously update current policies to respond to the dynamic situation of the global pandemic. Most especially, as our review has identified that the duration between infection episodes can be shorter than suggested in the 90 days regulation.

We found limited evidence on onwards transmission of recurrent SARS-CoV-2 infection. In total, only 15 studies assessed contact tracing or follow-up of re-positives, but four of these found evidence for onward transmission from re-positives [14,51,55,60]. This potential infectiousness in addition to the known reluctance of recovered patients to adhere to mitigation measures [15] emphasizes the need for further studies on onward transmission. Evidence from these studies can strongly impact on testing and tracing regulations, as well as on quarantine and isolation requirements.

As the pandemic progresses and as re-positive cases are reportedly increasing, it is essential to identify individuals who are at most risk of reinfection. In our review, we did not find conclusive evidence on risk factors, timing and mechanism of re-infection, nor a cause of re-positivity. In terms of risk factors, no reliable predictive marker was found, but prolonged initial illness [95], moderate disease severity, decreased leucocytes, low platelets and low CD4 count [96] were associated with re-positivity. The association of recurrent SARS-CoV-2 infection with age was controversial.

## Limitations

Synthesizing data from different study designs including preprints to respond to the pressing needs for scientific evidence, enabled us to provide robust evidence on the extent and characteristics of reinfections. However, there are some limitations to the review. Firstly, some included studies were lacking important information including details on timing of testing

and definition of re-infection. Critical appraisal was applied to take quality of studies into account. Secondly, restricting the language to English, could decrease generalizability of results as included studies may not cover all studies on recurrent SARS-CoV-2 infection. Thirdly, we did not perform double data extraction. But, in order to minimize data extraction error, a sample of 20% of extracted data was randomly cross checked. Finally, the RKI definition of reinfection probability has some limitations as confirmed infection considers viral load. But most included studies did not provide detailed information on the number of genetic copies of SARS-CoV-2 or perform virus cultures.

The present review has shown that re-positivity rates are high, but data on cause of re-positivity, infectivity and predictive markers are scarce. However, this review emphasizes the continuous need to update policies on contact tracing and quarantine regulations. Only by taking re-infections into account, it is possible to respond to the COVID-19 strategic preparedness and response plan of the WHO [102] and get in control of the global pandemic.

## Conclusion

In this review, we found that recurrence of a positive SARS-CoV-2 test among previously recovered cases is a commonly-reported phenomenon within the first few weeks from recovery. While some of these cases follow exposure, confirmed SARS-CoV-2 re-infections are rare. Fifty percent of genetically confirmed cases of re-infection were observed within 90 days after initial disease. Evidence on onwards transmission and predictive markers is limited but existent. With this high rate of recurrence of SARS-CoV-2, and mixed evidence of the risk to public health, policy makers need to re-consider current policies of contact tracing and quarantine regulations.

## Supporting information

**S1 PRISMA checklist. PRISMA 2020 checklist.**
(DOCX)

**S1 Table. Search strategy.**
(DOCX)

**S2 Table. Characteristics of SARS-COV-2 re-positive cases as reported in case studies.**
(DOCX)

**S3 Table. Critical appraisal of case reports included.**
(DOCX)

**S4 Table. Critical appraisal of case series included.**
(DOCX)

**S5 Table. Critical appraisal of observational studies included.**
(DOCX)

**S6 Table. Critical appraisal of reviews included.**
(DOCX)

## Acknowledgments

We would like to thank the team at RKI library for their support and for contribution to the search tools.

## Author Contributions

**Conceptualization:** Charbel El Bcheraoui.

**Data curation:** Seth Kofi Abrokwa, Sophie Alice Müller.

**Formal analysis:** Seth Kofi Abrokwa, Sophie Alice Müller, Alba Méndez-Brito.

**Investigation:** Seth Kofi Abrokwa, Sophie Alice Müller, Charbel El Bcheraoui.

**Methodology:** Seth Kofi Abrokwa, Sophie Alice Müller, Charbel El Bcheraoui.

**Project administration:** Charbel El Bcheraoui.

**Resources:** Seth Kofi Abrokwa, Sophie Alice Müller, Charbel El Bcheraoui.

**Supervision:** Johanna Hanefeld, Charbel El Bcheraoui.

**Validation:** Charbel El Bcheraoui.

**Visualization:** Seth Kofi Abrokwa, Sophie Alice Müller.

**Writing – original draft:** Seth Kofi Abrokwa, Sophie Alice Müller, Charbel El Bcheraoui.

**Writing – review & editing:** Seth Kofi Abrokwa, Sophie Alice Müller, Alba Méndez-Brito, Johanna Hanefeld, Charbel El Bcheraoui.

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
