## [Decision Letter · Decision Letter 0]

25 Aug 2021

PONE-D-21-17233

Recurrent SARS-CoV-2 infections and their potential risk to public health – A systematic review

PLOS ONE

Dear Dr. Charbel El Bcheraoui,

Thank you for submitting your manuscript to PLOS ONE. After careful consideration, we feel that it has merit but does not fully meet PLOS ONE’s publication criteria as it currently stands. Therefore, we invite you to submit a revised version of the manuscript that addresses the points raised during the review process.

We look forward to receiving your revised manuscript.

Kind regards,

Daniela Flavia Hozbor

Academic Editor

PLOS ONE

1. Please ensure that your manuscript meets PLOS ONE's style requirements, including those for file naming. The PLOS ONE style templates can be found at https://journals.plos.org/plosone/s/file?id=wjVg/PLOSOne_formatting_sample_main_body.pdf and https://journals.plos.org/plosone/s/file?id=ba62/PLOSOne_formatting_sample_title_authors_affiliations.pdf.

3. We note that this manuscript is a systematic review or meta-analysis; our author guidelines therefore require that you use PRISMA guidance to help improve reporting quality of this type of study. Please upload copies of the completed PRISMA checklist as Supporting Information with a file name “PRISMA checklist”.

Additional Editor Comments (if provided):

Reviewers' comments:

Reviewer's Responses to Questions

**Comments to the Author**

1. Is the manuscript technically sound, and do the data support the conclusions?

Reviewer #1: Yes

2. Has the statistical analysis been performed appropriately and rigorously? 

Reviewer #1: Yes

3. Have the authors made all data underlying the findings in their manuscript fully available?

Reviewer #1: Yes

4. Is the manuscript presented in an intelligible fashion and written in standard English?

Reviewer #1: Yes

5. Review Comments to the Author

Reviewer #1: While systematic reviews are quite important for clinical practice and helpful for adopting well-informed decisions/policies, they are less common in some nonexperimental situations, i.e., epidemiologic, or diagnostic studies, despite their relevance for appraising the quality of such studies. Within this context the study by Abrokwa et al. report results by systematically reviewing studies about the occurrence of a further positive SARS-CoV-2 test in people recovered from their COVID-19 episode. Their results point out that re-positivity rates in formerly recovered cases are worth considering, particularly in the ensuing weeks following recovery, highlighting the need to revise clinical and epidemiological strategies aimed at a better disease control.

Although systematic reviews are not invulnerable to the potential drawbacks of them, authors employed appropriate procedures to reduce the possibility of such inaccuracies in addition to explicitly describing the potential limitations of result interpretation (for instance studies published in English language).

Few comments follow:

Page 3, line 36: please update the number of licensed vaccines.

Page 3, lines 41-43: this statement must be rewritten considering current evidence about the durability of immunity (i.e., doi.org/10.1016/j.tim.2021.03.016;
doi.org/10.1016/j.eclinm.2021.100902;
doi.org/10.1038/s41586-021-03647-4;
doi.org/10.1016/j.chom.2021.04.015)

Page 11, line 155: the sum appears to be different from 1,100 re-positive cases.

6. PLOS authors have the option to publish the peer review history of their article (what does this mean?). If published, this will include your full peer review and any attached files.

Reviewer #1: No

---

## [Author Response · Author response to Decision Letter 0]

18 Nov 2021

Response to reviewers

Comment 1:

While systematic reviews are quite important for clinical practice and helpful for adopting well-informed decisions/policies, they are less common in some nonexperimental situations, i.e., epidemiologic, or diagnostic studies, despite their relevance for appraising the quality of such studies. Within this context the study by Abrokwa et al. report results by systematically reviewing studies about the occurrence of a further positive SARS-CoV-2 test in people recovered from their COVID-19 episode. Their results point out that re-positivity rates in formerly recovered cases are worth considering, particularly in the ensuing weeks following recovery, highlighting the need to revise clinical and epidemiological strategies aimed at a better disease control.

Although systematic reviews are not invulnerable to the potential drawbacks of them, authors employed appropriate procedures to reduce the possibility of such inaccuracies in addition to explicitly describing the potential limitations of result interpretation (for instance studies published in English language).

Response 1:

We thank the reviewer for the value they see in our paper. We hope the corrections meet their expectations. 

Comment 2:

Page 3, line 36: please update the number of licensed vaccines.

Response 2:

We thank the reviewer for point this out. Indeed, due to the increasing innovations to curb the pandemic, there has been significant additions to the COVID-19 vaccines to ensure global access in recent times. As recommended, we have change the number of licensed vaccines from three to six by reviewing recent COVID-19 vaccine report by the World Health Organization. 

Comment 3:

Page 3, lines 41-43: this statement must be rewritten considering current evidence about the durability of immunity i.e., doi.org/10.1016/j.tim.2021.03.016;
doi.org/10.1016/j.eclinm.2021.100902;
doi.org/10.1038/s41586-021-03647-4;
doi.org/10.1016/j.chom.2021.04.015)

Response 3:

We are grateful to the reviewer for pointing out the change in evidence on the immune response to post-natural COVID-19 infection and sharing insightful resources. We have examined the resources provided and have reviewed our initial statement on the immunological evidence based on recommendations from the reviewer as follows:

“Immunological evidence suggests that immune response to post-natural SARS-CoV-2 infection is transient with reported rapid depletion of antibodies in the first four months followed by gradual waning within a year.”

Comment 4:

Page 11, line 155: the sum appears to be different from 1,100 re-positive cases.

Response 4: 

We thank the reviewer for this comment. Indeed, if the total re-positive cases are added from the table, one would end up with 1144 cases. However, the last 44 cases are reported in a study without a denominator. For this reason, we reported the 1100 from the 180,185 separately from the 44 cases as follows:

“A total of 1,100 re-positives were reported out of 180,185 previously recovered patients, whereby one study reported 44 re-positives of an unknown total assessed”

Response to editor’s comments

Comment 1: We note that this manuscript is a systematic review or meta-analysis; our author guidelines therefore require that you use PRISMA guidance to help improve reporting quality of this type of study. Please upload copies of the completed PRISMA checklist as Supporting Information with a file name “PRISMA checklist”.

Response 1:

We thank the editor for reminding us of the PRISMA checklist. We have completed the PRISMA checklist as Supporting Information and uploaded it with our re-submission.

---

## [Editor Report · Decision Letter 1]

29 Nov 2021

Recurrent SARS-CoV-2 infections and their potential risk to public health – A systematic review

PONE-D-21-17233R1

Dear Dr.Charbel El Bcheraoui

We’re pleased to inform you that your manuscript has been judged scientifically suitable for publication and will be formally accepted for publication once it meets all outstanding technical requirements.

Kind regards,

Daniela Flavia Hozbor

Academic Editor

PLOS ONE
---

## [Editor Report · Acceptance letter]

2 Dec 2021

PONE-D-21-17233R1 

Recurrent SARS-CoV-2 infections and their potential risk to public health – A systematic review 

Dear Dr. El Bcheraoui:

I'm pleased to inform you that your manuscript has been deemed suitable for publication in PLOS ONE. Congratulations! Your manuscript is now with our production department. 

Kind regards, 

on behalf of

Dr. Daniela Flavia Hozbor 

Academic Editor

PLOS ONE